# Sex differences in associated factors for age-related hearing loss

Dong Woo Nam[iD][1], Min-Hyun Park[iD][2,3]☯*, Su Ji Jeong[iD][3], Kook Lae Lee[4,5], Ji Won Kim[4,5], Ji Bong Jeong[iD][5]☯*

1 Department of Otorhinolaryngology, Chungbuk National University Hospital, Cheongju, Republic of Korea, 2 Department of Otorhinolaryngology, Seoul National University College of Medicine, Seoul, Republic of Korea, 3 Department of Otorhinolaryngology, Boramae Medical Center, Seoul Metropolitan Government-Seoul National University, Seoul, Republic of Korea, 4 Department of Internal Medicine, Seoul National University College of Medicine, Seoul, Republic of Korea, 5 Department of Internal Medicine, Boramae Medical Center, Seoul Metropolitan Government-Seoul National University, Seoul, Republic of Korea

☯ These authors contributed equally to this work.
* jibjeong@snu.ac.kr (JBJ); drpark@snu.ac.kr (MHP)

**Data Availability Statement:** This study investigated the association between various factors and hearing loss. In addition to age and sex, it encompassed sensitive information, including baseline medical conditions, physical

## Abstract

The prevalence and age of onset of hearing loss differ according to sex. This study aimed to identify associated factors for age-related hearing loss (ARHL) and determine whether there are differences between males and females regarding associated factors for ARHL. This cross-sectional study used data from adults who underwent medical examinations including hearing tests from 2011 to 2021. A total of 2,349 individuals were included. The study conducted sex-specific analyses using both univariate and multiple regression. Univariate analysis employed logistic regression, while multiple regression involved variable selection through the augmented backward elimination method. Separate multiple logistic regression analyses were conducted for each sex. In the univariate analysis, among males, age, underweight, alcohol consumption, weight, and height exhibited statistical significance. Among females, age, hypertension, diabetes, dyslipidemia, obesity, sarcopenia, weight, height, age at menarche, and duration of hormone exposure were found to be significant factors. However, in the multiple logistic regression model for males, underweight, and smoking emerged as significant, while in females, age, weight, obesity, and age at menarche retained their significance. We found that there are different associated factors for ARHL in each sex. Assessment and counseling for smoking, obstetric history, underweight, and obesity may be beneficial in managing patients with ARHL.

## Introduction

Approximately 157 million people, accounting for 20.3% of the world's population, suffered from hearing loss in 2019, and this prevalence is expected to increase as the population ages [1]. Hearing loss causes difficulties in spoken communication, which negatively affects various aspects of an individual's life. Decreased cognitive function [2], loneliness [3], depression [4], and high unemployment rates [5] are observed in patients with hearing loss. It is almost impossible to improve hearing loss once it has progressed. Therefore, it is necessary to identify preventative measures. The most common risk factor of hearing loss is aging.

measurements such as height and weight, and obstetric history. While no personally identifiable information was included, the combination of this diverse information could potentially lead to individual identification. Therefore, in this study, we have chosen not to make the complete dataset publicly available. But the datasets generated during and/or analyzed during the current study are available and can be provided upon a reasonable request and with the approval of the IRB. For further inquiries or data requests, please directly contact the IRB of Boramae Medical Center at brmirb@gmail.com or 82-2-870-1851.

**Funding:** The author(s) received no specific funding for this work.

**Competing interests:** The authors have declared that no competing interests exist.

Age-related hearing loss (ARHL) is characterized by progressive, slow, persistent, bilateral, and symmetrical high-frequency hearing loss. Hearing deterioration may begin in individuals in their 20s and early 30s, although it usually occurs in people over the age of 50 years [6]. ARHL can be regarded as a multifactorial process of otological diseases that overlap with acquired hearing stress, trauma, and aging [7].

Chronic noise exposure and ototoxic drugs are well-known risk factors for hearing loss. Strong associations have been reported between a family history of hearing impairment and ARHL [8]. ARHL has been reported to be associated with Caucasian ethnicity [9] and lower socioeconomic status [10]. Associations between hearing loss and smoking [11], alcohol [12], and other chronic diseases, such as hypertension [13], diabetes [14], dyslipidemia [15], obesity [16], underweight [17], sarcopenia [18] and adiposity [19] have been reported. Short stature has been reported to be associated with a higher risk of hearing loss, potentially due to fetal growth issues involving insulin-like growth factor 1, which affects cochlear development [20].

In addition, there are differences in hearing loss according to sex: the prevalence is lower in females than in males [21] and the age at onset of hearing loss is later [22]. Research indicates a sex-specific association between factors related to abdominal fat and hearing loss, with males showing a correlation with high-frequency hearing loss, and females exhibiting a correlation with low-frequency hearing loss [23]. Research on risk factors for hearing loss has also noted sex-based distinctions, where, for instance, high triglyceride levels and a history of smoking have been identified as risk factors among males, while a high body mass index (BMI) has been associated with females [24].

In females, reproductive factors have been associated with various chronic conditions. For instance, associations have been reported between increased parity and breast cancer [25], age at menarche and cardiovascular diseases [26], as well as the duration of estrogen exposure and cardiovascular diseases [27]. However, research on the association between obstetric history and ARHL is currently limited. Due to the uncertainties related to onset and risk factors, we designed a large-scale retrospective study using health examination data. The purposes of this study were to identify associated factors for ARHL and determine whether there are differences between males and females in terms of associated factors for ARHL.

## Materials and methods

### Participants

This study used data from adults who underwent medical examinations at Boramae Medical Center from January 2011 to December 2021. Data collection began January 2022 and concluded July 2022. A total of 22,332 individuals who underwent medical examinations, including hearing tests, were screened. The context of their visit was primarily oriented toward healthcare and preventive healthcare objectives. Among them, 2,349 participants were analyzed, excluding those younger than 60, those with missing data, those with a history of occupational noise exposure, those with hearing loss other than ARHL, those with asymmetric hearing loss, and those with central nervous system diseases (Fig 1). All data obtained from participants were anonymized before statistical analysis. This study was approved by the Institutional Review Board of Boramae Medical Center (IRB no. 10-2021-143), and the requirement for informed consent was waived.

### Data collection

The participants fasted for 12 hours before visiting our medical center. Clinical information and test results were collected during the health examination. The height and weight of all patients were measured while standing. Body composition analysis was performed by a trained

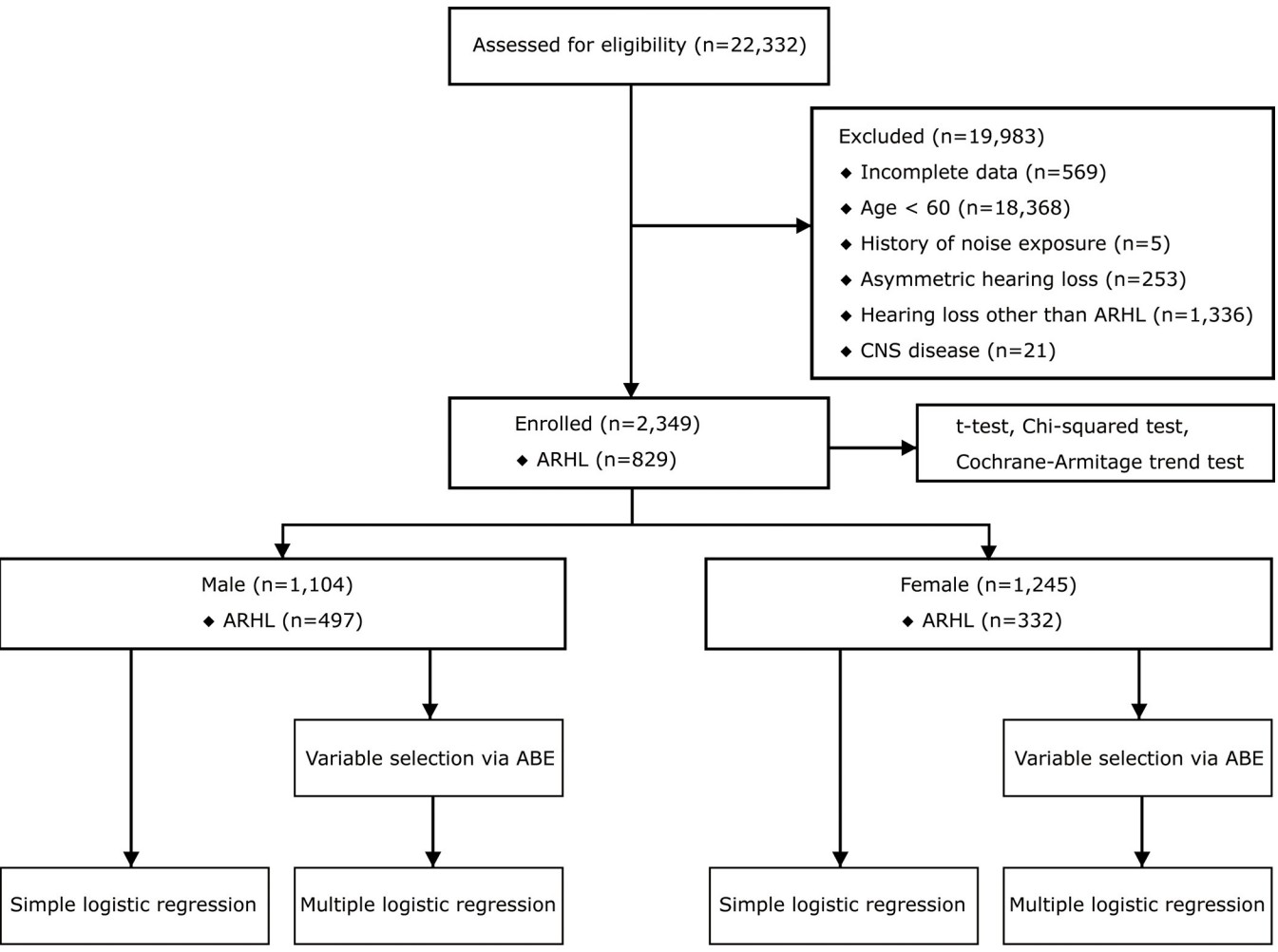

**Fig 1. CONSORT flow diagram.**

nurse using an InBody 720 (Biospace Co. Ltd., Seoul, Republic of Korea) following the manufacturer's protocol. appendicular skeletal muscle mass (ASM) and visceral fat area (VFA) were automatically calculated using the InBody 720. Clinical information, including age, sex, drinking, and smoking, was collected. For female participants, medical history, such as age at menarche, history of childbirth, age at menopause, and the use of hormone replacement therapy (HRT) was also investigated. Blood tests were performed, including evaluation of fasting glucose, glycated hemoglobin, high-density lipoprotein-cholesterol (HDL-C), and triglyceride levels [28]. Air conduction pure tone hearing thresholds were measured at 500, 1000, 2000, and 4000 Hz in both ears of the participants.

## Definitions

The average hearing threshold was calculated as the average of the 500, 1000, 2000, and 4000 Hz pure-tone hearing thresholds. Normal hearing was defined as all hearing thresholds at 500–4000 Hz were 25 dB or less in the better ear (the ear with the lower average hearing threshold). ARHL was defined as a 2 kHz threshold greater than 25 dB in the better ear and a 4 kHz threshold equal to or greater than the 2 kHz threshold in the same ear, which represents a modification of the criteria originally proposed by Sousa, C. S. et al [29]. Hearing loss that did

not fall into either of these categories was excluded from the analysis. Asymmetric hearing loss was defined as a difference of 20 dB or more in the average hearing threshold between the two ears and was excluded from the analysis [30]. Weight was measured in kg, and height was measured in cm. Obesity, underweight, visceral adiposity, and sarcopenia were used as the body composition parameters. Obesity was defined as BMI $\geq$ 25, while underweight was defined as BMI $<$ 18.5 [31]. Visceral adiposity was evaluated via VFA, measured using an InBody 720. Subjects whose VFA $\geq$ 100 $cm^2$ were categorized as part of the visceral adiposity group, while those with VFA $<$ 100 $cm^2$ were assigned to the normal group [32]. ASM was calculated as the sum of the lean skeletal muscle masses of the bilateral upper and lower limbs. Sarcopenia was evaluated using ASM% calculated as ASM $\times$ 100/weight. Sarcopenia was defined as having an ASM% below 29.0 in males and under 22.9 in females [33]. Hypertension was defined as systolic blood pressure $\geq$ 140 mmHg, diastolic blood pressure $\geq$ 90 mmHg, or the use of antihypertensive medications. Diabetes was defined as fasting glucose $\geq$ 126 mg/dL, glycated hemoglobin level $\geq$ 6.5%, or use of antidiabetic medications. Dyslipidemia was defined as triglyceride level $\geq$ 150 mg/dL, HDL-C $<$ 40 mg/dL in males and $<$ 50 mg/dL in females, or the use of anti-dyslipidemia medications. The duration of hormone exposure was defined as the period between menarche and menopause [27], as 0 for non-menarche, and as the period from menarche to the examination day for non-menopausal women.

## Statistical analyses

In demographics, categorical data, that is, sex, comorbidity, drinking, and smoking were expressed as frequency and ratio of the number of participants. Continuous data such as age, weight, height, and age at menarche were presented as the mean and standard deviation.

When comparing the characteristics of the normal hearing and ARHL groups, we used t-tests for continuous variables, Chi-squared tests for categorical variables, and Cochran Armitage trend test for smoking. For female participants, the two groups were compared on female-specific characteristics such as history of childbirth, age at menarche, and hormone exposure duration.

Sex-stratified analyses were conducted, employing both univariate and multiple regression techniques. Univariate regression analysis involved logistic regression, assessing ARHL in relation to each individual explanatory variable. Subsequently, multiple regression analysis was carried out exclusively on the variables chosen through the variable selection process.

Variable selection was performed using the augmented backward elimination (ABE) method, encompassing all potential variables for each sex and employing a significance level of 0.2, with a change-in-estimate threshold of 0.05 [34]. Multiple logistic regression analyses were separately performed for each sex, with a significance level set at 0.05 (Fig 1).

To assess multicollinearity in our regression models, we calculated the variance inflation factor (VIF). Initially, the VIF was computed for the multiple regression model incorporating all the considered variables. This was followed by computing the VIF for the simplified model, which was derived after the process of variable selection. The purpose of this analysis was to ensure that multicollinearity remains within acceptable limits, defined by having all coefficients in both models below the threshold value of 5.

The statistical package R (ver. 4.2.2, R Foundation for Statistical Computing, Vienna, Austria) was used for data analysis. In particular, the 'abe' package was used for ABE.

## Results

### Demographics

Of the 2,349 participants, 1,104 males (47%) and 1,245 females (53%) were included, with a mean age of 66.5 $\pm$ 5.6 years (range 60–94 years) (Table 1). Participants with ARHL were 829

**Table 1. Participants characteristics.**

| Variables | Total n = 2,349 | Normal hearing n = 1,520 | ARHL n = 829 | *P*-value |
|---|---|---|---|---|
| Age | 66.5±5.6 | 64.7±4.2 | 69.8±6.3 | < .001 |
| Sex | | | | < .001 |
| Male | 1,104 (47) | 607 (40) | 497 (60) | |
| Female | 1,245 (53) | 913 (60) | 332 (40) | |
| Hypertension | 1,198 (51) | 723 (48) | 475 (57) | < .001 |
| Diabetes | 445 (19) | 256 (17) | 189 (23) | < .001 |
| Dyslipidemia | 1,165 (50) | 740 (49) | 425 (51) | .249 |
| BMI | | | | < .001 |
| Normal | 1,470 (63) | 975 (64) | 495 (60) | |
| Obesity | 801 (34) | 509 (34) | 292 (35) | |
| Underweight | 78 (3.3) | 36 (2.4) | 42 (5.1) | |
| Visceral adiposity | 1001 (43) | 633 (42) | 368 (44) | .214 |
| Sarcopenia | 317 (14) | 176 (12) | 141 (17) | < .001 |
| Alcohol | 1,800 (77) | 1,165 (77) | 635 (77) | 1.000 |
| Smoking | | | | < .001 |
| Never | 1,511 (64) | 1,049 (69) | 462 (56) | |
| Former | 545 (23) | 321 (21) | 224 (27) | |
| Current | 293 (13) | 150 (9.9) | 143 (17) | |
| Weight (kg) | 61.6±10.4 | 61.5±10.2 | 61.7±10.6 | .611 |
| Height (cm) | 160.6±9.1 | 160.3±9.2 | 161.2±8.8 | .021 |

Abbreviations: ARHL, age-related hearing loss; BMI, body mass index.

(35%), and participants with hypertension, diabetes, dyslipidemia, obesity, underweight, visceral adiposity, and sarcopenia were 1,198 (51%), 445 (19%), 1,165 (50%), 801 (34%), 78 (3.3%), 1,001 (43%), and 317 (14%), respectively. There were 1,800 drinkers (77%), 293 current smokers (13%), 545 former smokers (23%), and 1,511 non-smokers (64%).

Among the 829 participants with ARHL, 497 were male (60%) and 332 were female (40%), and the mean age was 69.8 ± 6.3 years (range 60–94 years). Among the participants with ARHL, 475 (57%), 189 (23%), 425 (51%), 292 (35%), 42 (5.1%), 368(44%), and 141 (17%) had hypertension, diabetes, dyslipidemia, obesity, underweight, visceral adiposity and sarcopenia, respectively. Among the participants with ARHL, 635 were alcohol consumers (77%), 143 current smokers (17%), 224 former smokers (27%), and 462 non-smokers (56%) (Table 1).

There were significant differences in several characteristics between the ARHL group and the normal hearing group. The ARHL group exhibited greater age (t-test, $P < .001$), and taller height (t-test, $P = .021$), and the proportion of ARHL was higher in male, hypertension, diabetes, and sarcopenia (Chi-squared test, $P < .001$ each)). BMI was found to have a significant association with ARHL (Chi-squared test, $P < .001$), as well as smoking and ARHL (Cochran-Armitage trend test, $P < .001$). There was no significant difference in body weight (t-test, $P = .611$), dyslipidemia (Chi-squared test, $P = .249$), visceral adiposity (Chi-squared test, $P = .214$) and alcohol consumption (Chi-squared test, $P = 1.000$).

## Univariate logistic regression analysis by sex

Univariate logistic regression analysis of ARHL showed that in males, age ($P < .001$), underweight ($P = .001$), alcohol consumption ($P = .039$), weight ($P < .001$), and height ($P < .001$) were significant, and the odds ratios (ORs) for the variables except age and underweight were

**Table 2. Univariate logistic regression analysis of the association between age-related hearing loss and multiple explanatory variables, stratified by sex.**

| Variables | Male | | | Female | | |
|---|---|---|---|---|---|---|
| | OR | 95% CI | *P*-value | OR | 95% CI | *P*-value |
| Age | 1.209 | 1.175–1.245 | < .001 | 1.218 | 1.187–1.252 | < .001 |
| Hypertension | 1.119 | 0.881–1.422 | .358 | 1.745 | 1.355–2.251 | < .001 |
| Diabetes | 1.158 | 0.874–1.534 | .307 | 1.639 | 1.178–2.266 | .003 |
| Dyslipidemia | 1.016 | 0.800–1.289 | .897 | 1.465 | 1.135–1.894 | .003 |
| BMI | | | | | | |
| Normal | Reference | | | Reference | | |
| Obesity | 0.826 | 0.642–1.061 | .135 | 1.524 | 1.168–1.987 | .002 |
| Underweight | 3.557 | 1.710–8.119 | .001 | 1.806 | 0.919–3.421 | .076 |
| Visceral adiposity | 0.871 | 0.687–1.105 | .255 | 1.241 | 0.959–1.604 | .099 |
| Sarcopenia | 1.225 | 0.872–1.720 | .240 | 2.009 | 1.419–2.829 | < .001 |
| Alcohol | 0.684 | 0.476–0.980 | .039 | 0.809 | 0.623–1.054 | .115 |
| Smoking | | | | | | |
| Never | Reference | | | Reference | | |
| Former | 0.844 | 0.635–1.121 | .240 | 0.864 | 0.312–2.065 | .766 |
| Current | 1.241 | 0.894–1.726 | .198 | 0.821 | 0.183–2.703 | .758 |
| Weight (kg) | 0.973 | 0.960–0.985 | < .001 | 0.984 | 0.968–0.999 | .034 |
| Height (cm) | 0.956 | 0.935–0.976 | < .001 | 0.946 | 0.925–0.968 | < .001 |
| History of childbirth | | | | 1.019 | 0.612–1.765 | .943 |
| Age at menarche (years) | | | | 1.251 | 1.166–1.344 | < .001 |
| Hormone exposure duration (years) | | | | 0.953 | 0.931–0.975 | < .001 |
| HRT | | | | 0.949 | 0.660–1.345 | .773 |

Abbreviations: OR, odds ratio; CI, confidence interval; BMI, body mass index; HRT, hormone replacement therapy.

less than 1. In females, age (*P* < .001), hypertension (*P* < .001), diabetes (*P* = .003), dyslipidemia (*P* = .003), obesity (*P* = .002), sarcopenia (*P* < .001), age at menarche (*P* < .001) were significant with an OR greater than 1. Weight (*P* = .034), height (*P* < .001), hormone exposure duration (*P* < .001) had an OR less than 1 and were significant (Table 2).

## Multiple logistic regression analysis by sex

In males, the following variables were selected through the ABE method: age, height, BMI, and smoking status. After performing multiple logistic regression analysis with these variables, the ORs and 95% confidence intervals for each explanatory variable were 1.212 (1.178–1.250) for age, 0.977 (0.953–1.000) for height, 1.953 (1.342–2.854) for current smoking, 1.066 (0.770–1.480) for past smoking, 3.020 (1.350–7.311) for underweight, and 0.920 (0.694–1.218) for obesity.

In females, the following variables were selected: age, weight, height, hypertension, diabetes, BMI, and age at menarche. After performing multiple analysis with these variables, the OR and 95% confidence interval of each explanatory variable was 1.200 (1.167–1.236) for age, 0.953 (0.921–0.983) for weight, 1.016 (0.990–1.051) for height, 1.237 (0.906–1.684) for hypertension, 1.367 (0.854–2.166) for diabetes, 0.995 (0.412–2.309) for underweight, 2.102 (1.315–3.434) for obesity, and 1.119 (1.033–1.213) for age at menarche (Table 3).

When comparing the results to those of a univariate logistic regression analysis, it was observed that in males, smoking was not significant in the univariate analysis but became significant in the multiple analysis. Conversely, alcohol consumption, weight, and height were

**Table 3. Multiple logistic regression analysis of the association between age-related hearing loss and multiple explanatory variables, stratified by sex.**

| Variables | Male | | | Female | | |
|---|---|---|---|---|---|---|
| | OR | 95% CI | *P*-value | OR | 95% CI | *P*-value |
| Age (years) | 1.212 | 1.178–1.250 | < .001 | 1.200 | 1.167–1.236 | < .001 |
| Hypertension | | | | 1.237 | 0.906–1.684 | .179 |
| Diabetes | | | | 1.367 | 0.854–2.166 | .187 |
| BMI | | | | | | |
| Normal | Reference | | | Reference | | |
| Underweight | 3.020 | 1.350–7.311 | .010 | 0.995 | 0.412–2.309 | .990 |
| Obesity | 0.920 | 0.694–1.218 | .561 | 2.102 | 1.315–3.434 | .002 |
| Smoking | | | | | | |
| Never | Reference | | | Reference | | |
| Former | 1.066 | 0.770–1.480 | .700 | | | |
| Current | 1.953 | 1.342–2.854 | .001 | | | |
| Weight (kg) | | | | 0.953 | 0.921–0.983 | .004 |
| Height (cm) | 0.977 | 0.953–1.000 | .054 | 1.016 | 0.990–1.051 | .179 |
| Age at menarche (years) | | | | 1.119 | 1.033–1.213 | .006 |

Abbreviations: OR, odds ratio; CI, confidence interval; BMI, body mass index.

significant in the univariate analysis but were excluded in the multiple analysis. Among females, dyslipidemia, sarcopenia, weight, and hormone exposure duration were significant in the univariate analysis, but they were not included in the final model in the multiple analysis. Hypertension, diabetes, and height were included in the multiple analysis, but they were not found to be significant. Notably, in the case of height, the OR was less than 1 in the univariate analysis (0.946), but it was greater than 1 in the multiple model (1.016).

## Discussion

Although numerous factors associated with ARHL have been reported, in practice it is necessary to prioritize associated factors according to sex to help prevent ARHL. This retrospective study included a large number of participants and various covariates. Through ABE, it was identified that common associated factors for ARHL differ between sexes.

There are several methods for variable selection; backward elimination is appropriate for the prognostic model, and the ABE is suitable for the etiologic model [35]. As this was a study on the associated factors for ARHL, ABE was used.

Obesity was significant in the multiple analysis among females (*P* = .002), but was eliminated during the variable selection process for males. This outcome partially accounts for the inconsistency observed in prior studies regarding obesity as a contributing factor to ARHL. Inadequate adjustments for sex, weight, or smoking in previous research may have also contributed to this disparity. Notably, underweight was found to be significant in males, while low body weight was found to be significant in females, supporting earlier literature that identified underweight status as a risk factor for ARHL [17]. Smoking was significantly associated with ARHL in males. The OR increased for former (1.066) or current (1.953) smokers compared to those for never smokers, suggesting the dose-response relationship. But the OR in former smokers was not significant (*P* = .700). This is presumably due to the inhomogeneity of the participants of former smokers. Smoking, initially non-significant in univariate analysis, later proved significant in variable selection, highlighting the risk of excluding important variables when using univariate regression, consistent with prior research [36].

Unlike in the case of males, smoking was not included as a variable in the multiple logistic regression model for females. This result was based on the substantial disparity in the proportions of non-smokers between the two sex groups, with non-smokers accounting for 28% (304/1,104) of the male participants and 97% (1,207/1,245) of the female participants. As a result, the analysis pertaining to smokers was found to lack statistical significance and was consequently eliminated during the variable selection process.

Although many studies have been conducted on the difference between hearing loss in males and females, including the differences in associated factors for hearing loss according to sex [23, 24, 37], these studies were conducted with variables common to both males and females. The strength of this study is that it analyzed sex differences in ARHL by adding female-specific variables related to menarche and menopause, in addition to known associated factors for ARHL. To identify sex-specific associations, stratification, traditional product terms, and augmented product term methods have been proposed as research methods; in this study, the stratification method was selected to include female-specific variables.

There was a statistically significant difference in height between the ARHL and normal hearing groups, with the ARHL group being taller (t-test, $P = .021$, Table 1). However, subsequent univariate regression analysis by sex showed a statistically significant difference, with both males and females having an OR for height below 1 ($P < .001$ for each, Table 2). These results demonstrate Simpson's paradox, highlighting that associations may vary when examining the entire cohort and subgroups stratified by sex.

The results of both univariate and multiple analyses in this study exhibited inconsistencies. Variables that displayed significance in the univariate analysis, such as dyslipidemia, sarcopenia, weight, and duration of hormone exposure in females, were either excluded from the multiple analysis or lost their significance, as observed in the cases of hypertension, diabetes, and height. This inconsistency can be attributed to the strong association between each variable and age, which also exhibited a substantial correlation with ARHL. For instance, in females, sarcopenia demonstrated a significant negative association with ARHL in the univariate analysis (OR = 2.009, $P < .001$) but was subsequently eliminated in the multiple analysis. Notably, female participants with sarcopenia had an average age of 69.8 years, which was 3.7 years higher than female participants without sarcopenia. Given that the OR for age was estimated to be 1.2, the calculated OR for the age difference between the two groups was $1.2^{3.7} = 1.96$. Consequently, as the variance in ARHL between females with and without sarcopenia is primarily explained by age, sarcopenia was removed from the variable selection process. As exemplified in the case of age and sarcopenia mentioned above, when introducing multiple explanatory variables in multiple regression, it is imperative to consider multicollinearity.

In the multiple analysis for females, the OR for age at menarche yielded statistical significance, with an OR of 1.119 ($P = .006$). This finding may suggest a reduced risk of ARHL associated with early estrogen exposure, aligning with previous research indicating estrogen's protective role in hearing preservation [38, 39]. However, it's notable that while age at menarche was included as a variable, the duration of hormone exposure was excluded from the model. In our study, the duration of hormone exposure was defined as the timespan from menarche to menopause, irrespective of HRT utilization or its duration. In this study, we investigated the use of HRT, but we did not assess the duration of treatment, resulting in an incomplete analysis of its potential effects. Therefore, further comprehensive research is imperative to address this aspect comprehensively.

As this is a cross-sectional study, it is not possible to derive a causal relationship; therefore, further studies are required to confirm these findings. Additionally, the cross-sectional design of this study could have led to relatively elevated odds ratio values. Furthermore, owing to the limited sample size in this study, it's possible that only a subset of the variables considered

significant here might remain significant in a larger study. Consequently, these limitations should be taken into account when interpreting the study's results. At the time of data collection, the associated factors related to hearing loss, such as history of ototoxic drug exposure, family history of hearing loss, history of head trauma and infectious diseases, were not assessed. A tympanic examination was not performed, and only air conduction hearing thresholds were examined; therefore, conductive hearing loss was not excluded. Furthermore, an investigation into the history of sudden hearing loss was not conducted. Nonetheless, it is noteworthy that bilateral conductive or bilateral sudden hearing loss is uncommon among adults. Furthermore, the presence of unilateral conductive or unilateral sudden hearing loss did not impact the results, as the analysis focused on better ear hearing and excluded cases of asymmetric hearing loss. In addition, this study was conducted among individuals who voluntarily visited health check-up centers for the purposes of health management and prevention. As a result, the study may not be representative of the general population, and, consequently, there exists the potential for bias.

## Conclusions

Specific associated factors for ARHL in males include underweight and smoking. Specific associated factors for ARHL in females include obesity, low weight, and late menarche. Assessment and counseling regarding smoking, obstetric history, underweight, and obesity may be helpful in the management of patients with ARHL.

## Supporting information

**S1 Table. Detailed multiple logistic regression analysis (male).**
(PDF)

**S2 Table. Detailed multiple logistic regression analysis (female).**
(PDF)

## Author Contributions

**Conceptualization:** Min-Hyun Park, Ji Bong Jeong.

**Data curation:** Dong Woo Nam, Kook Lae Lee, Ji Won Kim, Ji Bong Jeong.

**Formal analysis:** Dong Woo Nam, Su Ji Jeong.

**Methodology:** Dong Woo Nam, Min-Hyun Park, Su Ji Jeong.

**Project administration:** Ji Bong Jeong.

**Visualization:** Dong Woo Nam.

**Writing – original draft:** Dong Woo Nam.

**Writing – review & editing:** Min-Hyun Park, Su Ji Jeong, Kook Lae Lee, Ji Won Kim, Ji Bong Jeong.

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
