## [Decision Letter · Decision Letter 0]

14 Sep 2023

PONE-D-23-24622Sex differences in risk factors for age-related hearing lossPLOS ONE

Dear Dr. Park,

Thank you for submitting your manuscript to PLOS ONE. After careful consideration, we feel that it has merit but does not fully meet PLOS ONE’s publication criteria as it currently stands. Therefore, we invite you to submit a revised version of the manuscript that addresses the points raised during the review process.

We look forward to receiving your revised manuscript.

Kind regards,

Paul Hinckley Delano, Ph.D.

Academic Editor

PLOS ONE

Journal Requirements:

Reviewers' comments:

Reviewer's Responses to Questions

**Comments to the Author**

1. Is the manuscript technically sound, and do the data support the conclusions?

Reviewer #1: Partly

Reviewer #2: Partly

2. Has the statistical analysis been performed appropriately and rigorously? 

Reviewer #1: No

Reviewer #2: No

3. Have the authors made all data underlying the findings in their manuscript fully available?

Reviewer #1: No

Reviewer #2: Yes

4. Is the manuscript presented in an intelligible fashion and written in standard English?

Reviewer #1: Yes

Reviewer #2: Yes

5. Review Comments to the Author

Reviewer #1: The “Sex differences in risk factors for age-related hearing loss” explores associated factors to hearing loss considering sex as qualitatively different phenomenon that can produce differences on risk factors. The study has a large sample size, and the main aim of the study is reasonable, and can be clinically useful. Despite its strengths, it has some caveats that should be considered by the authors.

1- Risk factors can’t be estimated from cross-sectional studies. This does not mean that this work is not a contribution to the field, but that authors should state that they are estimating associated factors, and discussing the potential bias derived from using this approach. It is critical to realize that odds ratio can vary substantially using cross-sectional vs longitudinal approaches.

2- Variable definitions are not supported in previous literature. Also, risk factors are categorical variables, but many of the predictors were entered as continues variables and no cut-off point was estimated. This is again aligned with an associated factor rationale rather than a risk factor.

3- The variables selected as potential risk factors should be properly addressed in the introduction. It is critical in this kind of study to present evidence supporting the link between the risk and the outcome to avoid spurious associations.

4- Similarly, variables such as menopause are included, without specifying how is entered (I assume is present, absent, however is not declared). This variable is an event, not an exposition. As such, is difficult to follow the rationale of considering it a risk factor. This rationale, as those in the other proposed variables, must be explicit and supported in previous literature.

5- It was not entirely clear the context at which patient arrive to consultation. Are they preventive, routine, or other? This may contribute to potential bias not discussed in the discussion.

6- Given the sex perspective of the study, a similar table to those reported in Table 1 should address descriptives by sex. For instance, many comments are given in the discussion of significant differences found in variables of Table 1, however, those differences are not divided by sex as logistic regressions.

7- The usual approach when estimating ORs in associated factors is to run simple logistic regression, not multivariate. The reason to do so is that risk factors are not independent, and this affects the estimation of ORs. For the case of this multivariate model, authors not only include a high number of predictors, which also negatively affects GLMs, but they also don’t check for collinearity. Authors should check variance inflation factor if this approach is used. Many variables are most likely highly correlated, and given the number of regressors, collinearity is most likely high.

8- Finally, considering the sample size, authors present high sensibility to detect minor but not relevant differences. This is not discussed. Only p-values are considered, without neglecting differences that may be considered as fairly irrelevant. Also, ORs estimated cross-sectionally, tend to present higher ORs. Considering these two elements, only a few results from the entire outcome would be promising for future longitudinal studies.

Reviewer #2: The article “Sex differences in risk factors for age-related hearing loss” is an interesting contribution to the knowledge of hearing loss in elderly adults, but some major modifications should be included before acceptance.

The main objective of the paper should be assessment of sex differences in risk factors for age related hearing loss (ARHL), since this is the tittle. Nevertheless, in the introduction there is only one phrase discussing this topic from previous publications. The authors should expand on this aspect so the reader can clearly understand the objective of the paper, or change the tittle.

The diagnosis of ARHL in the clinical setting usually relies on symmetric descending sensorineural hearing loss. Even though there is no clear age limit for the onset of this condition, most of the literature on the topic include adults over the age of 60 years old. In this paper, the age range spans between 40 and 94 years of age. Because of this, genetic-related adult-onset hearing loss could be included along with ARHL. My recommendation for the authors is to analyze the data again including only subjects older than 60 years of age. Also, there is no mention of asymmetric hearing loss. This is not typical of ARHL and these cases should be excluded.

There are some known risk factors for hearing loss in adults, such as head trauma or infectious deseases, that are not included. Please explain why.

Regarding hormonal exposure, there is no mention of menopausal hormonal replacement therapy. If there is no data this issue should be included in the discussion.

Table 1 should include intervals of confidence. Also I would recommend to change the % to make reference to the total according to hearing status and not to the total of subjects. For example, in sex distribution “Male” would be 51.1% in normal hearing and 64.1% in ARHL.

6. PLOS authors have the option to publish the peer review history of their article (what does this mean?). If published, this will include your full peer review and any attached files.

Reviewer #1: No

Reviewer #2: **Yes: **Mariela C. Torrente

---

## [Author Response · Author response to Decision Letter 0]

17 Nov 2023

Response to comment from the Reviewer #1:

Comment 1: 

Risk factors can’t be estimated from cross-sectional studies. This does not mean that this work is not a contribution to the field, but that authors should state that they are estimating associated factors, and discussing the potential bias derived from using this approach. It is critical to realize that odds ratio can vary substantially using cross-sectional vs longitudinal approaches.

Response: We followed your advice and changed all of the “risk factor” to “associated factor” in the body, including the title.

Sex differences in associated factors for age-related hearing loss

Comment 2: Variable definitions are not supported in previous literature. Also, risk factors are categorical variables, but many of the predictors were entered as continues variables and no cut-off point was estimated. This is again aligned with an associated factor rationale rather than a risk factor.

Response: We appreciate your feedback. We have provided references from existing literature to define age-related hearing loss, asymmetric hearing loss and obesity. And to redefine certain variables, specifically sarcopenia and visceral adiposity, we have employed cutoff values after consulting relevant literature. The script now reads (please refer to line 110-123):

ARHL was defined as a 2 kHz threshold greater than 25 dB in the better ear and a 4 kHz threshold equal to or greater than the 2 kHz threshold in the same ear, which represents a modification of the criteria originally proposed by Sousa, C. S. et al[18]. Hearing loss that did not fall into either of these categories was excluded from the analysis. Asymmetric hearing loss was defined as a difference of 20 dB or more in the average hearing threshold between the two ears and was excluded from the analysis[19]. Weight was measured in kg, and height was measured in cm. Obesity, underweight, visceral adiposity, and sarcopenia were used as the body composition parameters. Obesity was defined as BMI ≥ 25, while underweight was defined as BMI < 18.5[20]. Visceral adiposity was evaluated via VFA, measured using an InBody 720. Subjects whose VFA ≥ 100 ㎠ were categorized as part of the visceral adiposity group, while those with VFA < 100 ㎠ were assigned to the normal group[21]. ASM was calculated as the sum of the lean skeletal muscle masses of the bilateral upper and lower limbs. Sarcopenia was evaluated using ASM% calculated as ASM × 100/weight. Sarcopenia was defined as having an ASM% below 29.0 in males and under 22.9 in females[22]. 

Comment 3: The variables selected as potential risk factors should be properly addressed in the introduction. It is critical in this kind of study to present evidence supporting the link between the risk and the outcome to avoid spurious associations.

Response: We appreciate your suggestions, and as per your recommendations, we have improved the introduction and included references to the associated factors discussed in the main text. The script now reads (please refer to line 58-76):

Chronic noise exposure and ototoxic drugs are well-known risk factors for hearing loss. Strong associations have been reported between a family history of hearing impairment and ARHL[8]. ARHL has been reported to be associated with Caucasian ethnicity[9] and lower socioeconomic status[10]. Associations between hearing loss and smoking[11], alcohol[12], and other chronic diseases, such as hypertension[13], diabetes[14], dyslipidemia[15], obesity[16], underweight[17], and sarcopenia[18] have been reported. A higher risk of hearing loss has been reported for short stature[19] and adiposity[20].

In addition, there are differences in hearing loss according to sex: the prevalence is lower in females than in males[21] and the age at onset of hearing loss is later[22]. Research indicates a sex-specific association between factors related to abdominal fat and hearing loss, with males showing a correlation with high-frequency hearing loss, and females exhibiting a correlation with low-frequency hearing loss[23]. Research on risk factors for hearing loss has also noted sex-based distinctions, where, for instance, high triglyceride levels and a history of smoking have been identified as risk factors among males, while a high body mass index (BMI) has been associated with females[24].

In females, reproductive factors have been associated with various chronic conditions. For instance, associations have been reported between increased parity and breast cancer[25], age at menarche and cardiovascular diseases[26], as well as the duration of estrogen exposure and cardiovascular diseases[27]. However, research on the association between obstetric history and ARHL is currently limited.

Comment 4: Similarly, variables such as menopause are included, without specifying how is entered (I assume is present, absent, however is not declared). This variable is an event, not an exposition. As such, is difficult to follow the rationale of considering it a risk factor. This rationale, as those in the other proposed variables, must be explicit and supported in previous literature.

Response: We agree with your suggestion and have excluded menopause as an explanatory variable in subsequent analyses.

Comment 5: It was not entirely clear the context at which patient arrive to consultation. Are they preventive, routine, or other? This may contribute to potential bias not discussed in the discussion.

Response: We appreciate your feedback. The participants voluntarily sought healthcare services with a focus on health management and preventive purposes. This explanation has been added, and the corresponding limitations have been incorporated into the main text. The script now reads (please refer to line 83-85):

A total of 22,332 individuals who underwent medical examinations, including hearing tests, were screened. The context of their visit was primarily oriented toward healthcare and preventive healthcare objectives.

And the script now reads (please refer to line 290-294):

Furthermore, the presence of unilateral conductive or unilateral sudden hearing loss did not impact the results, as the analysis focused on better ear hearing and excluded cases of asymmetric hearing loss. In addition, this study was conducted among individuals who voluntarily visited health check-up centers for the purposes of health management and prevention. As a result, the study may not be representative of the general population, and, consequently, there exists the potential for bias.

Comment 6: Given the sex perspective of the study, a similar table to those reported in Table 1 should address descriptives by sex. For instance, many comments are given in the discussion of significant differences found in variables of Table 1, however, those differences are not divided by sex as logistic regressions.

Response: We appreciate your feedback. Simple logistic regression analyses were performed for each sex and are presented in Table 2.

Table 2. Univariate Logistic Regression Analysis of the Association between Age-Related Hearing Loss and Multiple Explanatory Variables, Stratified by Sex

Variables Male Female 

 OR 95% CI P-value OR 95% CI P-value

Age 1.209 1.175–1.245 <.001 1.218 1.187–1.252 <.001

Hypertension 1.119 0.881–1.422 .358 1.745 1.355–2.251 <.001

Diabetes 1.158 0.874–1.534 .307 1.639 1.178–2.266 .003

Dyslipidemia 1.016 0.800–1.289 .897 1.465 1.135–1.894 .003

BMI 

 Normal Reference Reference 

 Obesity 0.826 0.642–1.061 .135 1.524 1.168–1.987 .002

 Underweight 3.557 1.710–8.119 .001 1.806 0.919–3.421 .076

Visceral adiposity 0.871 0.687–1.105 .255 1.241 0.959–1.604 .099

Sarcopenia 1.225 0.872–1.720 .240 2.009 1.419–2.829 <.001

Alcohol 0.684 0.476–0.980 .039 0.809 0.623–1.054 .115

Smoking 

 Never Reference Reference 

 Former 0.844 0.635–1.121 .240 0.864 0.312–2.065 .766

 Current 1.241 0.894–1.726 .198 0.821 0.183–2.703 .758

Weight (kg) 0.973 0.960–0.985 <.001 0.984 0.968–0.999 .034

Height (cm) 0.956 0.935–0.976 <.001 0.946 0.925–0.968 <.001

History of childbirth 1.019 0.612–1.765 .943

Age at menarche (years) 1.251 1.166–1.344 <.001

Hormone exposure duration (years) 0.953 0.931–0.975 <.001

HRT 0.949 0.660–1.345 .773

Abbreviations: OR, odds ratio; CI, confidence interval; BMI, body mass index; HRT, hormone replacement therapy.

Comment 7: The usual approach when estimating ORs in associated factors is to run simple logistic regression, not multivariate. The reason to do so is that risk factors are not independent, and this affects the estimation of ORs. For the case of this multivariate model, authors not only include a high number of predictors, which also negatively affects GLMs, but they also don’t check for collinearity. Authors should check variance inflation factor if this approach is used. Many variables are most likely highly correlated, and given the number of regressors, collinearity is most likely high.

Response: We appreciate your feedback. It is not sufficient to estimate and interpret odds ratios from univariate analysis in the absence of controls. This is especially true when the relationship between an outcome and a risk factor is confounded by confounders that are not adequately controlled[1]. In such cases, the impact of the risk factor in question may be inaccurately assessed. This is why we conducted multiple logistic regression analyses in this study. 

For example, smoking was reported as a significant risk factor for males in the existing literature[2]. However, in this study, it was not significant in univariate analysis. Nevertheless, after selecting variables and adjusting for age, sex, and BMI, smoking was found to have a statistically significant relationship with age-related hearing loss.

Furthermore, when we examined multicollinearity, the Variance Inflation Factor (VIF) values were no higher than 10 in both the full model, which included all potential variables, and the simplified model obtained through variable selection.

We apologize for not fully addressing these points in the original manuscript and have added them to the text as you pointed out. The script now reads (please refer to line 230-232):

Smoking, initially non-significant in univariate analysis, later proved significant in variable selection, highlighting the risk of excluding important variables when using univariate regression, consistent with prior research[36].

And the script now reads (please refer to line 264-269):

As exemplified in the case of age and sarcopenia mentioned above, when introducing multiple explanatory variables in multiple regression, it is imperative to consider multicollinearity. In this study, variance inflation factor (VIF) was computed for both the multiple regression model utilizing all variables and the simplified model obtained after variable selection. For all coefficients in both models, VIF values did not exceed 10.

Comment 8: Finally, considering the sample size, authors present high sensibility to detect minor but not relevant differences. This is not discussed. Only p-values are considered, without neglecting differences that may be considered as fairly irrelevant. Also, ORs estimated cross-sectionally, tend to present higher ORs. Considering these two elements, only a few results from the entire outcome would be promising for future longitudinal studies.

Response: We concur with your feedback. This study is cross-sectional in nature, which may have led to relatively elevated odds ratio values. Furthermore, due to the limited sample size in this study, some of the variables that were found significant here may only be significant in certain large-scale studies. Therefore, when interpreting the results of this study, it is essential to consider these limitations. The script now reads (please refer to line 279-284):

As this is a cross-sectional study, it is not possible to derive a causal relationship; therefore, further studies are required to confirm these findings. Additionally, the cross-sectional design of this study could have led to relatively elevated odds ratio values. Furthermore, owing to the limited sample size in this study, it's possible that only a subset of the variables considered significant here might remain significant in a larger study. Consequently, these limitations should be taken into account when interpreting the study's results.

Response to comment from the Reviewer #2:

Comment 1: The main objective of the paper should be assessment of sex differences in risk factors for age related hearing loss (ARHL), since this is the tittle. Nevertheless, in the introduction there is only one phrase discussing this topic from previous publications. The authors should expand on this aspect so the reader can clearly understand the objective of the paper, or change the tittle.

Response: We appreciate your suggestions and have enhanced the introduction, incorporating additional literature to address sex differences in risk factors for hearing loss as per your recommendations. The script now reads (please refer to line 65-76):

In addition, there are differences in hearing loss according to sex: the prevalence is lower in females than in males[21] and the age at onset of hearing loss is later[22]. Research indicates a sex-specific association between factors related to abdominal fat and hearing loss, with males showing a correlation with high-frequency hearing loss, and females exhibiting a correlation with low-frequency hearing loss[23]. Research on risk factors for hearing loss has also noted sex-based distinctions, where, for instance, high triglyceride levels and a history of smoking have been identified as risk factors among males, while a high body mass index (BMI) has been associated with females[24].

In females, reproductive factors have been associated with various chronic conditions. For instance, associations have been reported between increased parity and breast cancer[25], age at menarche and cardiovascular diseases[26], as well as the duration of estrogen exposure and cardiovascular diseases[27]. However, research on the association between obstetric history and ARHL is currently limited.

Comment 2: The diagnosis of ARHL in the clinical setting usually relies on symmetric descending sensorineural hearing loss. Even though there is no clear age limit for the onset of this condition, most of the literature on the topic include adults over the age of 60 years old. In this paper, the age range spans between 40 and 94 years of age. Because of this, genetic-related adult-onset hearing loss could be included along with ARHL. My recommendation for the authors is to analyze the data again including only subjects older than 60 years of age. 

Response: We appreciate your feedback. We ran a new analysis restricting subjects to those aged 60 and older. The script now reads (please refer to line 86-88):

Among them, 2,349 participants were analyzed, excluding those younger than 60, those with missing data, those with a history of occupational noise exposure, those with hearing loss other than ARHL, those with asymmetric hearing loss, and those with central nervous system diseases (Fig 1). 

Comment 3: Also, there is no mention of asymmetric hearing loss. This is not typical of ARHL and these cases should be excluded.

Response: We appreciate your feedback. In this study, we referred to the criterion for asymmetric hearing loss proposed by Kurioka et al., which is "AC threshold between the right and left ear (RL-gap) >= 20 dB."[3] An explanation of this criterion has also been included in the main text. The script now reads (please refer to line 86-88):

Among them, 2,349 participants were analyzed, excluding those younger than 60, those with missing data, those with a history of occupational noise exposure, those with hearing loss other than ARHL, those with asymmetric hearing loss, and those with central nervous system diseases (Fig 1).

And the script now reads (please refer to line 114-115):

Asymmetric hearing loss was defined as a difference of 20 dB or more in the average hearing threshold between the two ears and was excluded from the analysis[30].

Comment 4: There are some known risk factors for hearing loss in adults, such as head trauma or infectious diseases, that are not included. Please explain why.

Response: We appreciate your feedback. This study utilized information from questionnaire items conducted during health check-ups, and we could not investigate the past history of head injuries or infectious diseases because these were not included in the questionnaire items. This constitutes a limitation of our study, and we have added a mention of this limitation. The script now reads (please refer to line 284-286):

At the time of data collection, the associated factors related to hearing loss, such as history of ototoxic drug exposure, family history of hearing loss, history of head trauma and infectious diseases, were not assessed.

Comment 5: Regarding hormonal exposure, there is no mention of menopausal hormonal replacement therapy. If there is no data this issue should be included in the discussion.

Response: We appreciate your feedback. The use of hormone replacement therapy in postmenopausal participants was investigated, and this has been included in the main text. However, the duration of hormone replacement therapy was not examined, which represents a limitation of the present study. The script now reads (please refer to line 273-278):

However, it's notable that while age at menarche was included as a variable, the duration of hormone exposure was excluded from the model. In our study, the duration of hormone exposure was defined as the timespan from menarche to menopause, irrespective of HRT utilization or its duration. In this study, we investigated the use of HRT, but we did not assess the duration of treatment, resulting in an incomplete analysis of its potential effects. Therefore, further comprehensive research is imperative to address this aspect comprehensively.

Comment 6: Table 1 should include intervals of confidence. Also I would recommend to change the % to make reference to the total according to hearing status and not to the total of subjects. For example, in sex distribution “Male” would be 51.1% in normal hearing and 64.1% in ARHL.

Response: We appreciate your feedback. To handle confidence intervals, we conducted univariate logistic regression analysis with stratification by sex, and the results have been summarized in Table 2. As you pointed out, we have made the necessary revisions to the notation in Table 1. 

Table 1. Participants characteristics

Variables Total

n=2,349 Normal hearing

n=1,520 ARHL

n=829 P-value

Age 66.5±5.6 64.7±4.2 69.8±6.3 <.001

Sex <.001

 Male 1,104 (47) 607 (40) 497 (60) 

 Female 1,245 (53) 913 (60) 332 (40) 

Hypertension 1,198 (51) 723 (48) 475 (57) <.001

Diabetes 445 (19) 256 (17) 189 (23) <.001

Dyslipidemia 1,165 (50) 740 (49) 425 (51) .249

BMI <.001

 Normal 1,470 (63) 975 (64) 495 (60) 

 Obesity 801 (34) 509 (34) 292 (35) 

 Underweight 78 (3.3) 36 (2.4) 42 (5.1) 

Visceral adiposity 1001 (43) 633 (42) 368 (44) .214

Sarcopenia 317 (14) 176 (12) 141 (17) <.001

Alcohol 1,800 (77) 1,165 (77) 635 (77) 1.000

Smoking <.001

 Never 1,511 (64) 1,049 (69) 462 (56) 

 Former 545 (23) 321 (21) 224 (27) 

 Current 293 (13) 150 (9.9) 143 (17) 

Weight (kg) 61.6±10.4 61.5±10.2 61.7±10.6 .611

Height (cm) 160.6±9.1 160.3±9.2 161.2±8.8 .021

Abbreviations: ARHL, age-related hearing loss; BMI, body mass index

References

1. Sun GW, Shook TL, Kay GL. Inappropriate use of bivariable analysis to screen risk factors for use in multivariable analysis. J Clin Epidemiol. 1996;49(8):907-16.

2. Kumar A, Gulati R, Singhal S, Hasan A, Khan A. The effect of smoking on the hearing status-a hospital based study. J Clin Diagn Res. 2013;7(2):210-4.

3. Kurioka T, Sano H, Furuki S, Yamashita T. Speech discrimination impairment of the worse-hearing ear in asymmetric hearing loss. Int J Audiol. 2021;60(1):54-9.

---

## [Decision Letter · Decision Letter 1]

3 Jan 2024

PONE-D-23-24622R1Sex differences in associated factors for age-related hearing lossPLOS ONE

Dear Dr. Park,

Thank you for submitting your manuscript to PLOS ONE. After careful consideration, we feel that it has merit but does not fully meet PLOS ONE’s publication criteria as it currently stands. Therefore, we invite you to submit a revised version of the manuscript that addresses the points raised during the review process. Respond to reviewer 1 minor concerns.

We look forward to receiving your revised manuscript.

Kind regards,

Paul Hinckley Delano, Ph.D.

Academic Editor

PLOS ONE

Journal Requirements:

Reviewers' comments:

Reviewer's Responses to Questions

**Comments to the Author**

1. If the authors have adequately addressed your comments raised in a previous round of review and you feel that this manuscript is now acceptable for publication, you may indicate that here to bypass the “Comments to the Author” section, enter your conflict of interest statement in the “Confidential to Editor” section, and submit your "Accept" recommendation.

Reviewer #1: All comments have been addressed

Reviewer #3: All comments have been addressed

2. Is the manuscript technically sound, and do the data support the conclusions?

Reviewer #1: Yes

Reviewer #3: Yes

3. Has the statistical analysis been performed appropriately and rigorously? 

Reviewer #1: Yes

Reviewer #3: Yes

4. Have the authors made all data underlying the findings in their manuscript fully available?

Reviewer #1: Yes

Reviewer #3: Yes

5. Is the manuscript presented in an intelligible fashion and written in standard English?

Reviewer #1: Yes

Reviewer #3: Yes

6. Review Comments to the Author

Reviewer #1: The draft has improved substantially and most of my observations were solved. I only have minor comments that I left below.

From line 115 to 125 seems to be a change in format.

In line 142 an space is missing after "elimination"

Regarding my comment 8, maybe I was not clear. The problem is not that you have a limited sample size, quite the opposite. Statistics tend to produce more false positive results when having around sample sizes of 800 or above. Therefore, this study can easily detect significant results which are false positives or true positives with low effect sizes. The inclusion of Cis helps to better interpret the results. Therefore, I would consider that currently this is a solved issue.

I understand the use of BMI, but not so clear heigh. Even weight alone is easy to infer why is entered in the model. But heigh alone does not make much sense, and there is no mention to it in the introduction explaining a potential causal role in audition loss. I suggest the authors include information in the introduction regarding height or removing it.

Finally, please provide a reference for VIF cut off =10. That is a rather high cut off value (removing high should reduce VIF values anyway). Also, this information should be provided in methods rather than discussion.

Reviewer #3: The authors have conducted a cross-sectional study with a large number of individuals, in which they identified associated factors for age-related hearing loss and then separated them into men and women. The authors re-analysed these data, incorporating new tables and statistical models for correct interpretation and from that, new conclusions. They also took into account the limitations of the study itself (in relation to its conclusions) that, being a cross-sectional study, it is generally not possible to establish a causal relationship between the different factors. Finally, the authors responded to, incorporated and reanalysed in a good way what was requested by both reviewers throughout this paper.

From the above, I believe that this paper is fit for publication. For my part, I only have reservations about the conclusion which, in general, is very similar to the three previous lines before the discussion section (I suggest deleting lines 294 to 296 from the paper)

7. PLOS authors have the option to publish the peer review history of their article (what does this mean?). If published, this will include your full peer review and any attached files.

Reviewer #1: No

Reviewer #3: No

---

## [Author Response · Author response to Decision Letter 1]

14 Jan 2024

Response to comment from the Reviewer #1:

Comment 1:

From line 115 to 125 seems to be a change in format.

In line 142 an space is missing after "elimination"

Response: We appreciate your feedback. The formatting issue you highlighted has been addressed. The script now reads (please refer to line 117-127):

Obesity was defined as BMI ≥ 25, while underweight was defined as BMI < 18.5. Visceral adiposity was evaluated via VFA, measured using an InBody 720. Subjects whose VFA ≥ 100 〖cm〗^2 were categorized as part of the visceral adiposity group, while those with VFA < 100 〖cm〗^2 were assigned to the normal group. ASM was calculated as the sum of the lean skeletal muscle masses of the bilateral upper and lower limbs. Sarcopenia was evaluated using ASM% calculated as ASM × 100/weight. Sarcopenia was defined as having an ASM% below 29.0 in males and under 22.9 in females. Hypertension was defined as systolic blood pressure ≥ 140 mmHg, diastolic blood pressure ≥ 90 mmHg, or the use of antihypertensive medications. Diabetes was defined as fasting glucose ≥ 126 mg/dL, glycated hemoglobin level ≥ 6.5%, or use of antidiabetic medications. Dyslipidemia was defined as triglyceride level ≥ 150 mg/dL, HDL-C < 40 mg/dL in males and < 50 mg/dL in females, or the use of anti-dyslipidemia medications.

And we have also added the missing spaces as indicated. The script now reads (please refer to line 143):

Variable selection was performed using the augmented backward elimination (ABE) method,

Comment 2:

I understand the use of BMI, but not so clear heigh. Even weight alone is easy to infer why is entered in the model. But heigh alone does not make much sense, and there is no mention to it in the introduction explaining a potential causal role in audition loss. I suggest the authors include information in the introduction regarding height or removing it.

Response: We sincerely appreciate your valuable feedback. We acknowledge that our initial manuscript did not adequately address the relationship between height and hearing loss. In response to your suggestion, we have revised the Introduction section to include a reference to previous study that explore this relationship. Additionally, we have elaborated on the authors' hypothesis concerning this correlation. We believe these amendments will significantly strengthen our manuscript and thank you for your insightful suggestions. The script now reads (please refer to line 61-64):

Associations between hearing loss and smoking[11], alcohol[12], and other chronic diseases, such as hypertension[13], diabetes[14], dyslipidemia[15], obesity[16], underweight[17], sarcopenia[18] and adiposity[19] have been reported. Short stature has been reported to be associated with a higher risk of hearing loss, potentially due to fetal growth issues involving insulin-like growth factor 1, which affects cochlear development[20].

Comment 3:

Finally, please provide a reference for VIF cut off =10. That is a rather high cut off value (removing high should reduce VIF values anyway). Also, this information should be provided in methods rather than discussion.

Response: We are grateful for your insightful feedback. We concur with your observation that a threshold of 10 for the Variance Inflation Factor (VIF) is excessively high to effectively rule out multicollinearity. Upon re-evaluating our models, including the variable 'height,' we found that the highest VIF value recorded is 4.6 for 'weight.' This value is well within the widely accepted threshold of 5 for VIF values, indicating a lower risk of multicollinearity.

Furthermore, we agree with your recommendation that this analysis should be included in the Methods section rather than the Discussion. We have revised our manuscript to reflect this change, ensuring that our approach to addressing multicollinearity is clearly outlined in the appropriate section. Thank you for guiding these improvements in our manuscript. The script now reads (please refer to line 147-152):

To assess multicollinearity in our regression models, we calculated the variance inflation factor (VIF). Initially, the VIF was computed for the multiple regression model incorporating all the considered variables. This was followed by computing the VIF for the simplified model, which was derived after the process of variable selection. The purpose of this analysis was to ensure that multicollinearity remains within acceptable limits, defined by having all coefficients in both models below the threshold value of 5.

Response to comment from the Reviewer #3:

I only have reservations about the conclusion which, in general, is very similar to the three previous lines before the discussion section (I suggest deleting lines 294 to 296 from the paper)

Response: We appreciate your feedback. We've revised the manuscript as you pointed out.

---

## [Editor Report · Decision Letter 2]

17 Jan 2024

Sex differences in associated factors for age-related hearing loss

PONE-D-23-24622R2

Dear Dr. Park,

We’re pleased to inform you that your manuscript has been judged scientifically suitable for publication and will be formally accepted for publication once it meets all outstanding technical requirements.

Kind regards,

Paul Hinckley Delano, Ph.D.

Academic Editor

PLOS ONE
---

## [Editor Report · Acceptance letter]

13 Feb 2024

PONE-D-23-24622R2 

PLOS ONE

Dear Dr. Park, 

I'm pleased to inform you that your manuscript has been deemed suitable for publication in PLOS ONE. Congratulations! Your manuscript is now being handed over to our production team.

Kind regards, 

on behalf of

Dr. Paul Hinckley Delano 

Academic Editor

PLOS ONE